# The Influence of Hydrothermal Aging on the Dynamic Friction Model of Rubber Seals

**DOI:** 10.3390/polym12010102

**Published:** 2020-01-04

**Authors:** Jian Wu, Hang Luo, Haohao Li, Benlong Su, Youshan Wang, Zhe Li

**Affiliations:** Center for Rubber Composite Materials and Structures, Harbin Institute of Technology, Weihai 264209, China; luohanghit@163.com (H.L.); lhh9403@163.com (H.L.); rains853@163.com (B.S.); lizhe0223@sina.com (Z.L.)

**Keywords:** polymer, friction model, cylinder, hydrothermal aging, rubber

## Abstract

Cylinder has become an indispensable and important pneumatic actuator in the development of green production technology. The sealing performance of the cylinder directly affects its safety and reliability. Under the service environment of the cylinder, hydrothermal aging of the rubber sealing ring directly affects the dynamic friction performance of the cylinder. So, the dynamic friction model of the cylinder has been developed based on the LuGre friction model, which considers the influence of hydrothermal aging. Here, the influences of the static friction coefficient and Coulomb friction coefficient on the friction model are analyzed. Then, the aging characteristic equation of rubber is embedded in the model for revealing the influence of aging on the friction coefficient of the model. Results show that the aging temperature, aging time, and compressive stress affects the friction coefficient; the variation of the static friction coefficient is larger than that of the Coulomb friction coefficient. The improved cylinder friction model can describe the influence of the aging process on the cylinder friction characteristics, which is of great significance in the design of the cylinder’s dynamic performance.

## 1. Introduction

Pneumatic technology is wildly used in industry due to its high speed, safety, and cleanness. Although the cylinder has a wide range of applications, the cylinder itself is not suitable for accurate control due to its poor stiffness, strong non-linearity, and the disadvantages of temperature and air pressure. The sealing performance of the cylinder is very important to the normal operation of the whole mechanism and its safety and reliability [1]. The friction characteristics of the cylinder, especially the friction characteristics under different operating conditions, is one of the main factors that make it difficult to accurately control the cylinder. In recent years, the friction model of the cylinder has been developed and used to control the cylinder. However, friction models are also affected by the seal ring aging, which is directly related to the dynamic friction performance of the cylinder. So, it is of great significance to study the influence of hydrothermal aging on the dynamic friction model of rubber seals, which can lay the foundation for improving the performance and service life of the cylinder as well as the sustainable development of the economy.

Friction is a widespread and very complex phenomenon [2], especially in soft material friction. A mathematical model has been widely developed to study the friction process based on a lot of friction tests [3,4]. Friction force is not only related to surface roughness, but also to the lubrication state [5]. The static friction coefficient is not a constant [6], and it is higher than that of dynamic friction, which is related to the initial residence time and starting speed of rubber specimens in the friction surface [7]. Friction characteristics of the cylinder are very important, which is related to the friction model [8,9]. However, one of the most important causes of the complexity in pneumatic cylinder control is the friction force between piston and rubber seals, as friction force is highly nonlinear, which causes the well-known stick-slip phenomena. Several friction models have been developed. It is well-known that static friction models, such as the Karnopp model [10], are incapable of correctly reproducing the phenomena. The friction force of cylinder dependence on the previously followed trajectory and on the time moving parts are previously still [11]. The Dahl model [12] is not directly applicable to pneumatic cylinders since it does not include the viscous phenomenon. The LuGre friction model represents a good trade-off between complexity and accuracy for fitting the friction properties of cylinders [13,14,15]—a typical LuGre friction model is shown in Figure 1.

Besides, the influence of parameters of the LuGre friction model on the pneumatic system was investigated for the gravity compensation cylinder [16]. The LuGre friction model between rubber rollers and paper was also developed to study the effect of the viscous damping coefficient on the model [17]. The LuGre model was used to describe the friction phenomenon for the friction compensation of rodless cylinders [18]. Compared to the experimental results, it was found that the LuGre model cannot simulate the actual friction characteristics of the hydraulic actuator, so a modified LuGre model was proposed to simulate the actual dynamic characteristics of friction for hydraulic actuators with high accuracy [19]. It was also found that the LuGre model could not fully characterize the friction characteristics of hydraulic cylinders; thus, an improved LuGre model was developed for correcting the viscous friction and dynamic characteristics of sliding oil film, which was better than the traditional model [20,21].

As above, the LuGre model has been widely used in practical engineering applications. However, rubber seal aging has a great effect on the friction process of the cylinder under the service environment. So, the rubber seals of the cylinder are taken as the research object, then the rubber aging characteristic equation is established. Based on the LuGre model, the improved cylinder friction model is developed, which is of great significance to understand the friction characteristics change in the service process of the rubber seals. It lays the foundation for the safe service and accurate control of the cylinder under actual working conditions.

## 2. Dynamic Friction Model of the Cylinder

### 2.1. LuGre Model

The LuGre model simplifies the rough protruding contact on the friction surface into elastic bristles, which have a certain stiffness and damping, and the stiffness of bristles on the friction lower surface is obviously greater than that on the upper surface. Under the action of tangential force between the friction surfaces, the elastic bristles on the upper surface deform and produce a friction force that impedes the movement of the friction surfaces. When the elastic bristle deforms to a certain extent, the friction surface slides relatively (seen in Figure 2).

In the LuGre friction model, the average deformation of bristle is expressed by *z*, and the friction force and friction state are expressed by [11]:(1)$$F_{f}=\sigma_{0}z+\sigma_{1}\frac{dz}{dt}+\sigma_{2}\nu$$
(2)$$\frac{dz}{dt}=\nu -\frac{\left|\nu \right|}{g(u)}z$$
(3)$$g(u)=\frac{1}{\sigma_{0}}\left[F_{c}+(F_{s}-F_{c})e^{-{\left(\frac{\nu }{\nu_{s}}\right)}^{2}}\right]$$
where *v* is relative velocity between two surfaces, *z* is average elastic shape variable, *F_f_* is the friction force, *F_c_* the is Coulomb friction force, *F_s_* is the static friction force, *v_s_* is the Stribeck speed, *g*(*u*) is a function describing the Stribeck speed, $\sigma_{0}$ the is stiffness coefficient, $\sigma_{1}$ is the damping coefficient, and $\sigma_{2}$ is coefficient of viscosity friction.

When the friction velocity is constant, the expression of the steady friction force of the LuGre model can be given by:(4)$$F_{f}=\left[F_{c}+(F_{s}-F_{c})e^{-{\left(\frac{v}{v_{s}}\right)}^{2}}\right]\mathrm{sgn}(v)+\sigma_{2}v_{r}$$

### 2.2. Improved Friction Model

Considering that the shape of the cylinder seals is variable and the contact state is complex, the traditional LuGre model is not completely applicable to the analysis of the friction characteristics of the rubber seals. A distributed LuGre friction model can be used to develop the friction model of the rubber seals with considering the contact distribution.

Here, in the friction model of seal in contact with the cylinder wall, the contact region can be assumed as the contact width *L* (seen in Figure 3). Considered the contact part is composed of a tiny unit, and the contact pressure is different, we can deduce the contact pressure and friction force of each unit. Similarly, the contact pressure between the rubber seal and the cylinder wall can be obtained by integrating the contact pressure of the unit.

The mathematical model is given by:(5)$$\frac{dz(t,\xi )}{dt}=v_{r}-\frac{v_{r}}{g(v_{r})}*z(t,\xi )$$
(6)$$F(t)=\sigma_{0}z+\sigma_{1}\dot{z}+\sigma_{2}v_{r}$$
(7)$$g(v_{r})=\frac{1}{\sigma_{0}}\left[\mu_{c}+(\mu_{s}-\mu_{c})e^{-{\left(\frac{v}{v_{s}}\right)}^{2}}\right]\int_{0}^{l}f_{n}(\zeta )d(\xi )$$
where *z* is the average elastic shape variable, *F* is the friction force of sealing rings, *f_n_* is the pressure distribution function of sealing ring contact area, $\mu_{c}$ is the Coulomb friction coefficient, and $\mu_{s}$ is the static friction coefficient. $\mu_{c}$ is usually thought to be less than $\mu_{s}$.

The distributed model can be assumed by the following conditions:(1)Although the load is not uniformly distributed on the whole contact surface, different load distribution equations can be selected according to the contact states of different sealing rings and cylinders.(2)The sliding velocity between each element and the friction surface is the same, and its velocity is equal to *v*.

The cylinder contact pressure distribution functions are given by:(8)$$\mathrm{Uniform}\;\mathrm{distribution}:\;f_{n}(\zeta )=\frac{F_{n}}{L}$$
(9)Sine distribution: fn(ζ)=πFn2Lsin(πζL)
(10)Exponential distribution: fn(ζ)=e−λ(ζL)fn(0),λ≥0
(11)Parabolic distribution: fn(ζ)=3Fn2L[1−(ζ−L2L2)2]

Considered the contact characteristics, the LuGre model can be extrapolated to its steady-state model. When the cylinder moves with a uniform speed, the average elastic shape variable in the LuGre model basically remains unchanged; that is, when the model becomes steady-state, *v* is a constant (dz(t,ζ)/dt=0), then Equation (5) is set to 0, and *z* can be expressed by:(12)$$z(t,\zeta )=g(v_{r})\mathrm{sgn}(v)$$

When Equation (7) is substituted to Equation (12), *z* can be given by:(13)$$z(t,\zeta )=g(v_{r})=\frac{1}{\sigma_{0}}\left[\mu_{c}+(\mu_{s}-\mu_{c})e^{-{\left(\frac{v}{v_{s}}\right)}^{2}}\right]\mathrm{sgn}(v)\int_{0}^{l}f_{n}(\zeta )d(\zeta )$$

Then, when it is substituted to Equation (6), the steady friction force can be given by:(14)$$F(t)=\{\left[\mu_{c}+(\mu_{s}-\mu_{c})e^{-{\left(\frac{v}{v_{s}}\right)}^{2}}\right]\mathrm{sgn}(v)\}\int f_{n}(\zeta )d(\zeta )+\sigma_{2}v_{r}$$

Then, the model static parameter of the rubber seal can be fitted according to Equation (14).

Considered the effect of aging, the fitting model of the rubber friction coefficient and aging time can be given by:(15)$$\mu^{\prime }=\mu_{0}-a\ln(t+b)-c$$
where *μ′* is friction coefficient; $\mu_{0}$ is initial friction coefficient; *a*, b, and c are aging factors; and t is aging time.

### 2.3. Effect of Model Parameters

There are four static parameters in the friction model. Here, $\sigma_{2}$ is related to lubricating oil, and $\mu_{c}$ and $\mu_{s}$ are related to the materials and properties of the friction pair. In generally, the cylinder friction model is analyzed due to the change of $\mu_{c}$ and $\mu_{s}$, which lays the foundation for parameter identification.

#### 2.3.1. Static Friction Coefficient

In the friction model, $\mu_{c}$ is 0.3, the contact pressure is 50 N, $\sigma_{2}$ is 0.2, and $\upsilon_{s}$ is 5 mm/s. When the static friction coefficient increases from 0.25 to 0.35, the influence of the static friction coefficient on the friction model is shown in Figure 4. It can be seen that the change of the static friction coefficient has a significant impact on the cylinder friction model. When the static friction coefficient decreases from 0.35 to 0.25, the initial static friction force decreases correspondingly; however, it has no impact on the Stribeck speed. The results also indicate that the static friction coefficient mainly affects the cylinder friction model at low speed, but it has no effect on the static friction coefficient at high speed.

#### 2.3.2. Coulomb Friction Coefficient

Figure 5 shows the effect of Coulomb friction coefficient on the friction model under the same condition in Figure 4.

When the Coulomb friction coefficient decreases from 0.30 to 0.20, the initial static friction force remains basically unchanged, but the friction force decreases correspondingly after the Stribeck velocity. It also indicates that with the increase of the Coulomb friction coefficient, the friction force increases at high speed, that is, the Coulomb friction coefficient mainly affects the cylinder friction model at high speed.

From the above, the influence of the static friction coefficient and the Coulomb friction coefficient on different speeds can be determined, which lay the foundation for parameter identification of the friction model.

## 3. Experimental Setup

In order to find the relationship between the rubber friction force and the speed after aging, rubber aging and friction coefficient tests were carried out to obtain the experimental data for the cylinder friction model.

### 3.1. Friction Test

In friction tests, rubber is applied to a load of 5 kg, the lead screw is controlled to drive the bottom-feeding platform to move, and the friction force is tested under different feed speed conditions.

The friction test platform of the sealing material has been developed for observation of friction characteristics. It consists of a normal loading part, a horizontal feed part, a measurement control system, and a tooling fixture, etc. Figure 6 shows a schematic diagram of the friction test structure.

The normal loading part is loaded with weights, and the loading process is controlled by the air cylinder. The sealing rubber sample is fixed in the tooling fixture. The horizontal feed part is comprised of a servo drive motor and the sideway; the lateral friction force is obtained by the force sensor. The force and displacement are measured by the data acquisition card and real-time data acquisition. The developed test device is shown in Figure 7, and each test was done three times. Besides, the accuracy of the force sensor is ±0.2 N.

### 3.2. Hydrothermal Aging Test

Based on ISO 188-2011, aging tests of rubber have been carried out in the homemade hydrothermal aging test chamber under the conditions of the compression state (compressing the test pieces to 80%, 90%, and uncompressed state of the original length), aging temperature (20° C, 40 °C, 60 °C, and 80 °C), and aging times (1 day, 2 days, 4 days, 8 days, 12 days, and 16 days).

## 4. Result and Discussion

### 4.1. Compression Ratio

The friction test has been carried out on the rubber specimens under different compression conditions, the predicted and measured friction force of the rubber specimens under different compression conditions, and the temperature of 60 °C are shown in Figure 8. Figure 9 shows the Coulomb friction coefficient and the static friction coefficient after aging.

It can be seen that when the pre-compression increases, the friction coefficient decreases. In the uncompressed state, the static friction coefficient of the rubber sealing material is 0.835, and the coefficient of static friction reduces to 0.822 under the compression ratio of 80%.

### 4.2. Temperature

Figure 10 shows the friction force under the compression of 80%, the temperature of 20 °C, 40 °C, 60 °C, and 80 °C. Then, the friction curve is fitted by the steady-state friction model. Besides, the Coulomb friction coefficient and static friction coefficient under different temperature and aging time are shown in Figure 11.

When the aging temperature increases, the friction coefficient decreases. Here, the static friction coefficient is decreased from 0.935 to 0.855 when the temperature increases from 40 °C to 80 °C under the 2 days of aging. It is the same as the Coulomb friction coefficient. From Figure 11, it can be seen that the variation ranges of the static friction coefficient are larger than the Coulomb friction coefficient, and the aging temperature is more likely to affect the friction coefficient.

### 4.3. Aging Time

Figure 12 shows the friction force of the rubber under the temperature of 60 °C, the compression of 80%, and the aging times of 1, 2, 4, 8, 12, and 16 days. Then, the friction curve is fitted to achieve the parameters of the steady-state friction model. Figure 13 shows the Coulomb friction coefficient and the static friction coefficient under different aging times.

Results indicate that the aging time has a great influence on the friction force of the rubber, especially in the first 4 days, which is similar to the rubber aging rate. In the first 4 days, the static friction coefficient decreases from 0.93513 to 0.75859, and the Coulomb friction coefficient decreases from 0.609 to 0.48429. From 4 to 16 days, there is no obvious change; here, the static friction coefficient decreases from 0.75859 to 0.70319, and the Coulomb friction coefficient decreases from 0.48429 to 0.44781.

### 4.4. Effect of Aging on Friction Model

Aging has a great effect on the mechanical properties of the rubber sealing ring, which leads to the changes in the friction coefficient. Figure 14 shows the relation curve between friction force and velocity of rubber after 16 days of aging under the temperature of 80 °C and the compression of 80%. It can be seen that the maximum friction value decreases from 49.4 N to 35.1 N after 16 days of aging under the temperature of 80 °C. Therefore, it is very important to study the effect of aging on friction model parameters.

The friction coefficient of rubber decreases with the aging time, and it can be found that the aging time has the same effect on the static friction coefficient and the Coulomb friction coefficient. Therefore, the relation curve of rubber friction coefficient and aging time is shown in Figure 13.

Based on Equation (15), parameters of the Coulomb friction coefficient can be obtained by: a_1_ = 0.03307, b_1_ = 0.1324, c_1_ = 0.06621. Similarly, parameters of the static friction coefficient can be given by: a_2_ = 0.04448, b_2_ = 0.0804, c_2_ = 0.11159. By comparing the static friction coefficient and the Coulomb friction coefficient of the model with the test data, it can be found that the error of the fitting results is less than 5%.

Based on the predicted model, the Coulomb friction coefficient after 16 days of aging is 0.48082, and the static friction coefficient is 0.7. The cylinder friction model is shown in Figure 15 for fitting and comparison with experimental data. It can be seen that the model can fit the cylinder friction and velocity curves better.

## 5. Conclusions

With the extensive application of the cylinder, the aging of rubber seals brings a severe challenge to mechanical performance. Therefore, the typical cylinder seal is taken as the research object, and the dynamic friction behavior of the cylinder in the aging process has been studied based on the improved LuGre model. The main conclusions are given by:(1)The steady friction model of the cylinder is developed based on the LuGre model. Results show that the model can well describe the friction process of the cylinder in the aging process.(2)The influence of aging temperatures, time, and compression conditions on the friction force have been revealed. The friction coefficient decreases with the increasement of aging temperature, aging time, and compression. The effect on static friction coefficient is larger than that of the Coulomb friction coefficient.(3)The effect of aging on the cylinder friction model parameters has been investigated. The parameters of the Coulomb friction coefficient and static friction coefficient have been fitted.

## Figures and Tables

**Figure 1 polymers-12-00102-f001:**
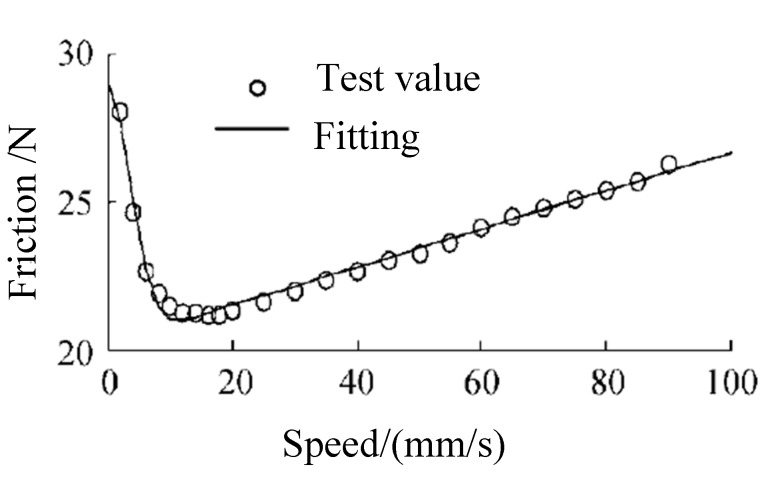
Fitting curve of the LuGre friction model [14].

**Figure 2 polymers-12-00102-f002:**
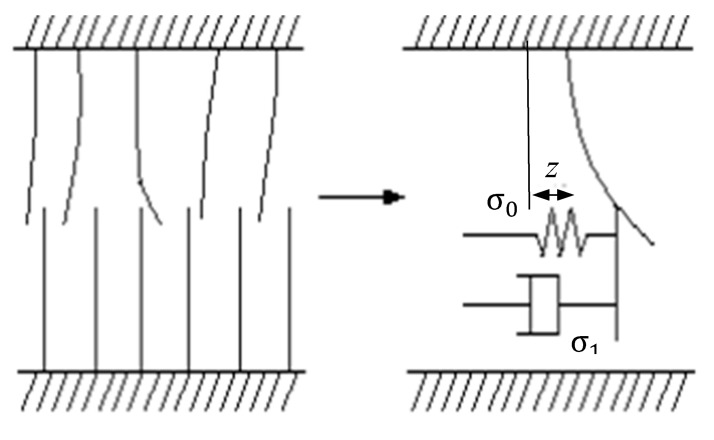
The LuGre friction model.

**Figure 3 polymers-12-00102-f003:**
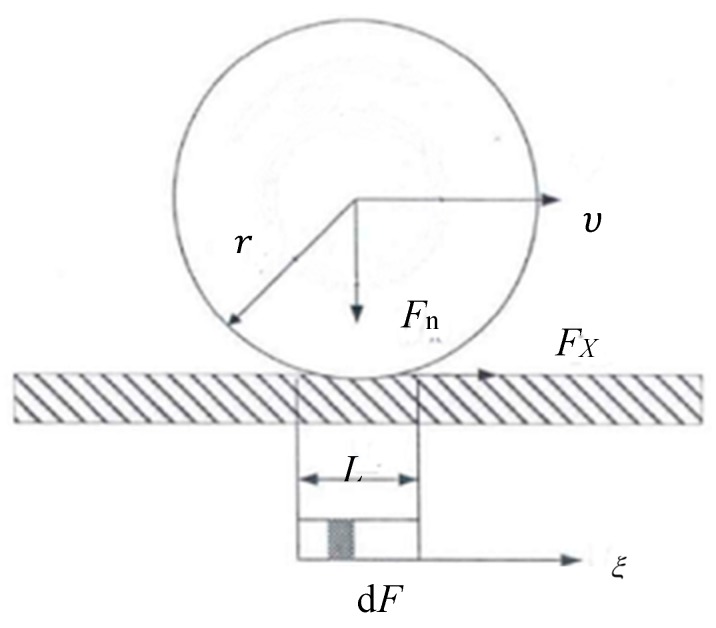
Friction model considered the contact distribution.

**Figure 4 polymers-12-00102-f004:**
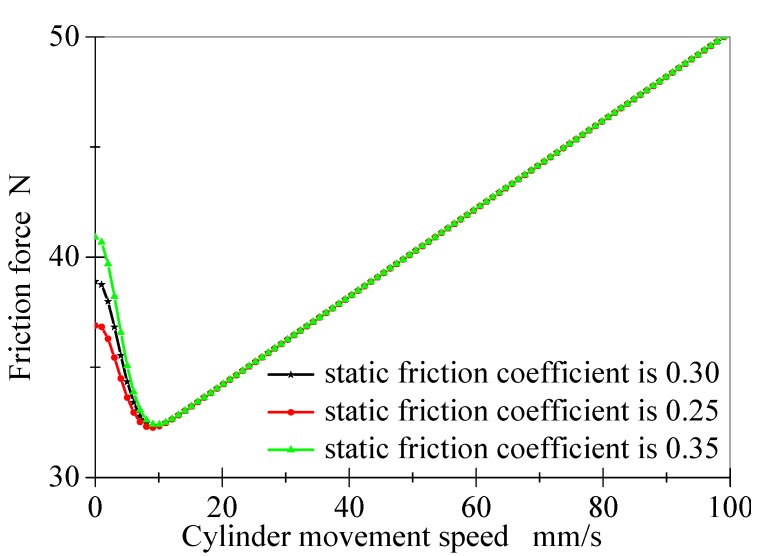
Influence of friction coefficient change on the LuGre friction model.

**Figure 5 polymers-12-00102-f005:**
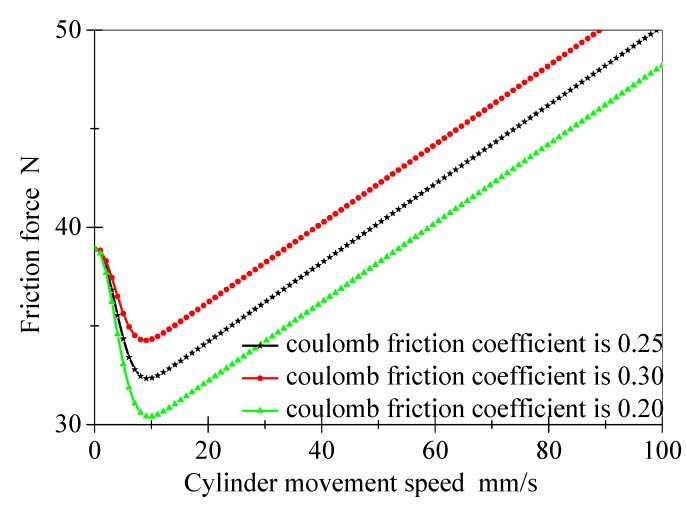
The influence of friction coefficient changes on cylinder friction model.

**Figure 6 polymers-12-00102-f006:**
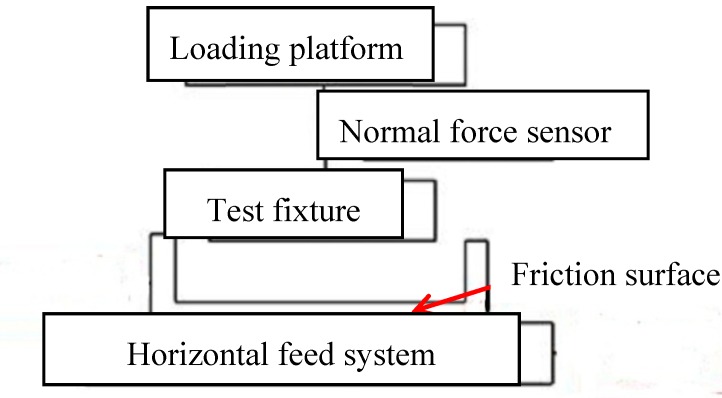
Schematic diagram of friction test structure.

**Figure 7 polymers-12-00102-f007:**
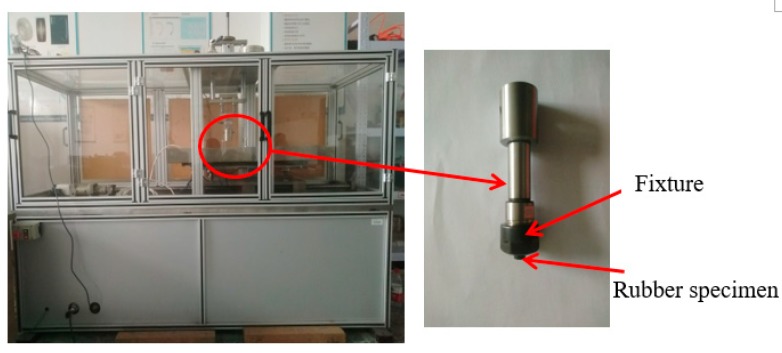
Friction test equipment.

**Figure 8 polymers-12-00102-f008:**
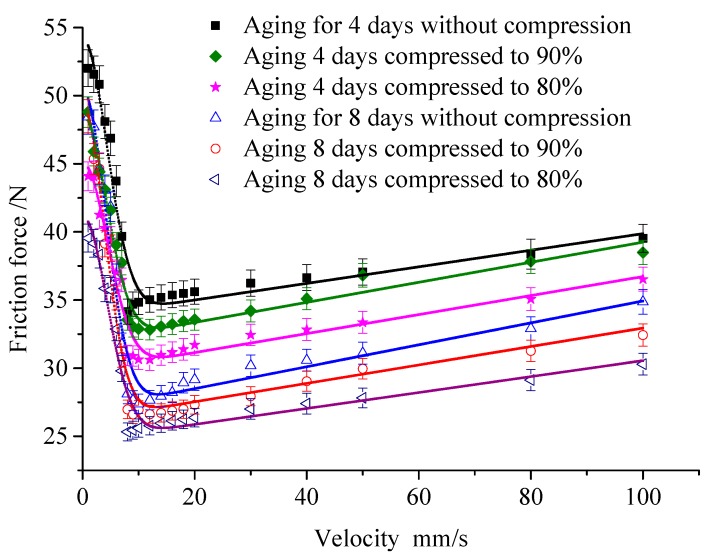
Effect of compression ratio on friction.

**Figure 9 polymers-12-00102-f009:**
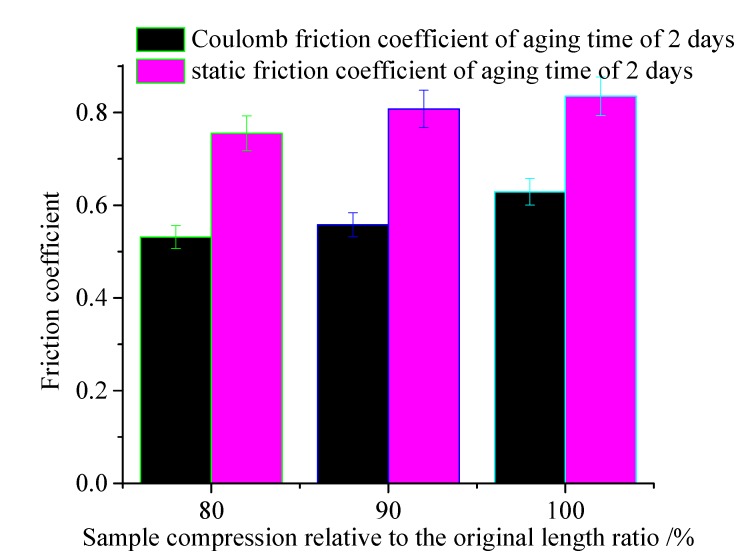
Friction coefficient of test specimens under different compression ratio.

**Figure 10 polymers-12-00102-f010:**
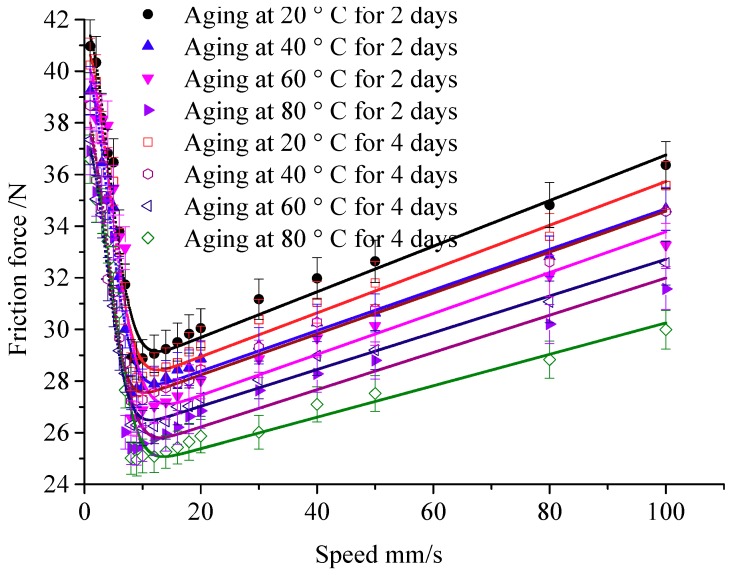
Effect of aging temperature on friction force.

**Figure 11 polymers-12-00102-f011:**
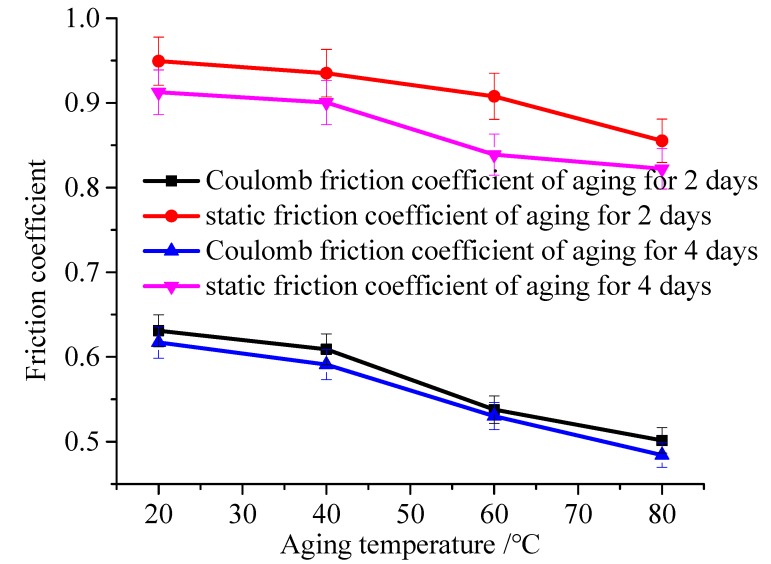
Friction coefficients under different temperature.

**Figure 12 polymers-12-00102-f012:**
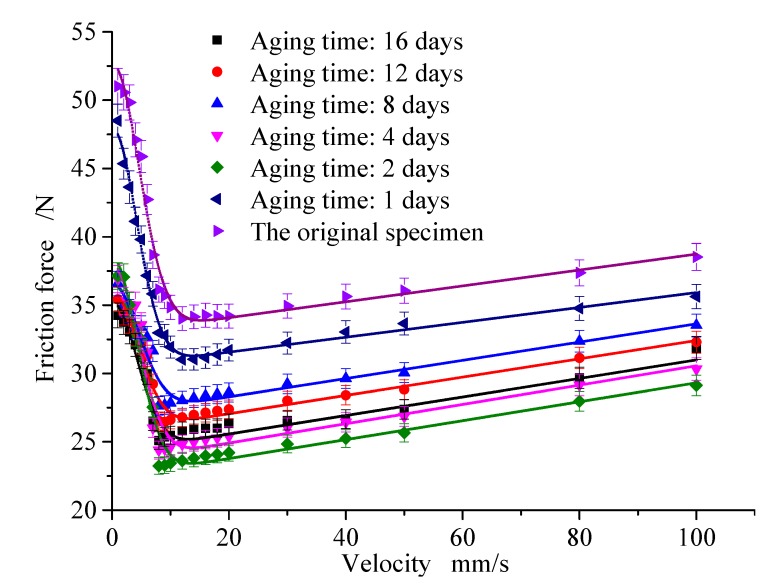
Effect of aging time.

**Figure 13 polymers-12-00102-f013:**
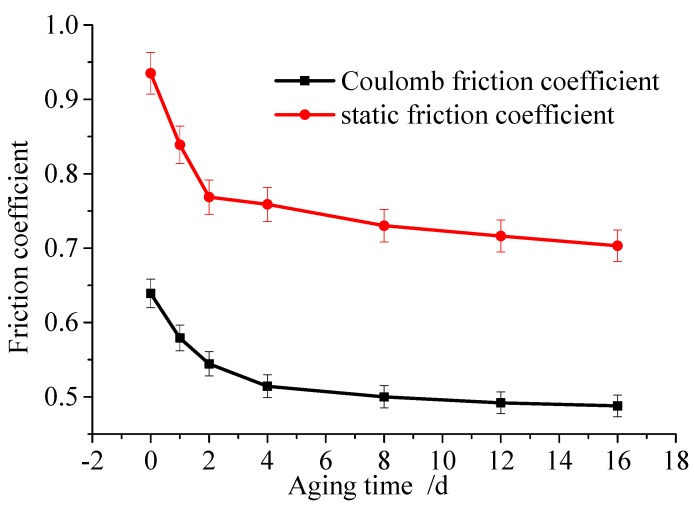
Test curves and friction coefficients of specimens with different aging time.

**Figure 14 polymers-12-00102-f014:**
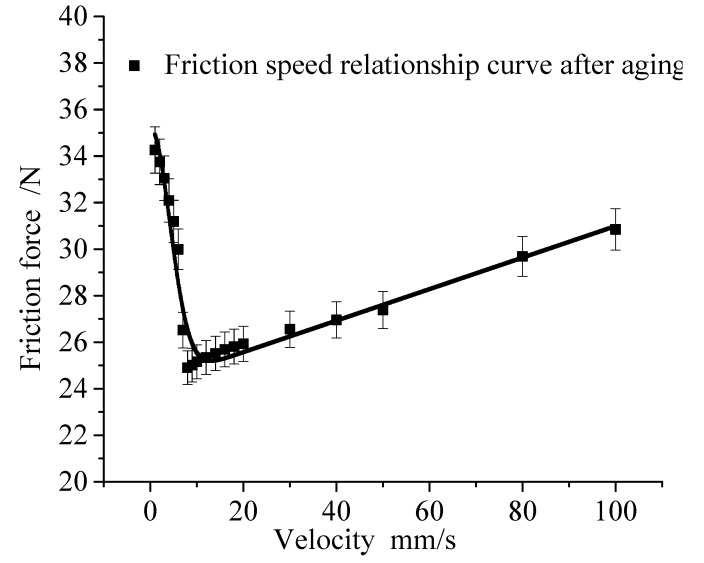
Friction model before and after aging.

**Figure 15 polymers-12-00102-f015:**
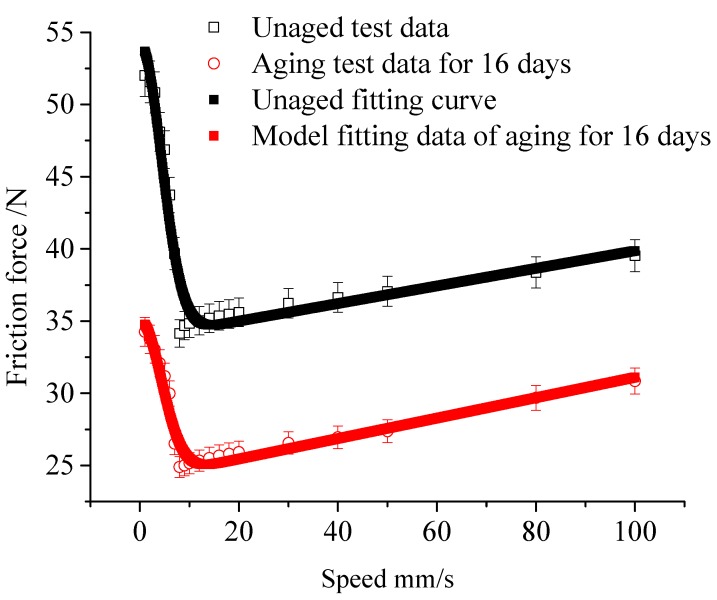
Comparison between aging prediction model and experimental test results.

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
