# Peer review of "The Influence of Hydrothermal Aging on the Dynamic Friction Model of Rubber Seals"

_polymers, 2020, doi:10.3390/polym12010102_

Round 1

Reviewer 1 Report

The introduction must reflect the problem that the LuGre model improves versus other models.

The experimental procedure must be fully described

A precise figure of the coordinate systems must be included (Eq. 1)

The application of this type of rubber is cylindrical, and there is no evidence of the model in three dimensions.

The friction coefficient that the authors proposed is not validated.

In every figure for experimental results, an error estimation must be presented

They must compare the results with different models

Author Response

Please find enclosed A REVISED electronic version of the paper polymers-650086 entitled:  “The influence of hydrothermal aging on the Dynamic Friction model of Rubber seals”, authored by Wu Jian, Luo Hang, Li Haohao, Su Benlong, Wang Youshan, Li zhe.

We thank very much the reviewers for the general overall comments on the paper.

The point-by-point response to the reviewer’s comments can be seen in the attachment file. 

Reviewer 2 Report

English grammar and syntax has to be checked carefully throughout the manuscript. There are several spelling mistakes in the manuscript. Paragraph 1 of the Introduction is redundant as all the people are very much aware of these basics. I guess the authors should directly start discussing about the problem on hand, which is about the sealing in a cylinder. I suggest that a proper background about the sealing in a cylinder and the challenges faced which are unsolved and how the present study is going to address this challenge should be presented in a proper manner, which is missing in the manuscript in the present form. Section2: Is a bit confusing. Did the authors use the two models to evaluate the COF for the cylinder at different speeds or are they presenting the data which is already published. This should be clarified. I did not see any Experimental methodology before the discussion of the results which confuses the readers. An extensive experimental methodology should be presented. The authors did not specify as to how many times did they conduct the experiments for the repeatability of the results. If they have done it more than once then there should be error bars indicated in the figures, example in Figure 6. I guess, without the experimental methodology, it is difficult to understand the experimental results.

Author Response

(The authors gave the same response as above.)

Round 2

Reviewer 1 Report

Authors have responded to all the comments

Author Response

Dear Reviewer

Please find enclosed A REVISED electronic version of the paper polymers-650086 entitled:  “The influence of hydrothermal aging on the Dynamic Friction model of Rubber seals”, authored by Wu Jian, Luo Hang, Li Haohao, Su Benlong, Wang Youshan, Li zhe.

We thank very much the reviewers for the general overall comments on the paper. 

For what concerns REVIEWER1 comments: 

Comments and Authors answers

Authors have responded to all the comments.

Thanks for your comments. Besides, we have improved the English language all the paper.

Reviewer 2 Report

The authors have added the experimental methodology, but it is still lacking a comprehensive description of all the characterization techniques used, their procedures well described etc. So, I suggest the authors to relook at this section. The authors have still not answered the question as to how many times the experiments were repeated for repeatability. They have added the error bars to the graphs, but I doubt the correctness of these bars as for all the data points the error bars seems to be the same, which cannot be correct. Please check.

Author Response

Dear Reviewer

Please find enclosed A REVISED electronic version of the paper polymers-650086 entitled:  “The influence of hydrothermal aging on the Dynamic Friction model of Rubber seals”, authored by Wu Jian, Luo Hang, Li Haohao, Su Benlong, Wang Youshan, Li zhe.

We thank very much the reviewers for the general overall comments on the paper. 

For what concerns REVIEWER2 comments: 
Comments and Authors answers

The authors have added the experimental methodology, but it is still lacking a comprehensive description of all the characterization techniques used, their procedures well described etc. So, I suggest the authors to relook at this section. The authors have still not answered the question as to how many times the experiments were repeated for repeatability. They have added the error bars to the graphs, but I doubt the correctness of these bars as for all the data points the error bars seems to be the same, which cannot be correct. Please check.

Thanks for pointing out this. We have revised the section according to the commands.

The section has been revised by:

“The friction test platform of the sealing material has been developed for observation of friction characteristics. It consists of a normal loading part, a horizontal feed part, a measurement control system and a tooling fixture, etc. Fig. 6 shows a schematic diagram of friction test structure.

The normal loading part is loaded with weights, and the loading process is controlled by the air cylinder. The sealing rubber sample is fixed in the tooling fixture. The horizontal feed part is comprised of a servo drive motor and the sideway, the lateral friction force is obtained by the force sensor. The force and displacement is measured by the data acquisition card and real-time data acquisition. The developed test device is shown in Fig.7, and each test done 3 times. Besides, accuracy of the force sensor is ±0.2N.”

The experiments were repeated for 3 times. The data used in the article is the average of each test.

We have checked the error bars, the error bars is not the same, which is from 3%-7%。Here, the test accuracy of the friction test equipment is high (accuracy of the force sensor is ±0.2N) and the repeatability is good. Besides, the main reason is that the data range is large in the figures, so the error bars is relatively small and it seems to be the same.

Round 3

Reviewer 2 Report

The authors have replied to all my queries.